# Production Challenges in Least Developed Countries

**Matthias Brönner \*** , **Skander Salah and Markus Lienkamp**

Institute of Automotive Technology, Technical University of Munich, Boltzmannstraße 15,
85748 Garching, Germany; skander.salah@tum.de (S.S.); Lienkamp@ftm.mw.tum.de (M.L.)
**\*** Correspondence: broenner@ftm.mw.tum.de; Tel.: +49-89-289-15907

**Abstract:** Local production sites in least developed countries offer sustainability for both multinational corporations and local society. However, corporations often hesitate because of uncertain environmental influences on production sites in these countries. To minimize planning uncertainties, we aim to identify and categorize the challenges of local production in least developed countries. Therefore, we conduct a research on local production challenges described in current literature. Our results indicate that the influences can be clustered and occur independent of the country. To show practical relevance and actuality of the identified production challenges, we conducted two case studies. Additionally, these studies give examples for organizational, product-specific and technological solutions to overcome the prevailing challenges. In summary, we support the removal of barriers that keep corporates from setting up local production sites in least developed countries.

**Keywords:** production; least developed countries; environmental conditions; Africa

## 1. Introduction

In least developed countries a connection between rising social standards and economic growth exists ([1] p. 53). Also, a correlation between demand for manual labor and rising income is confirmed ([2] p. 48). Studies demonstrate that the automotive industry in particular offers great potential for least developed countries through its far-reaching value chains [3]. Nevertheless, there is great uncertainty among companies whether they should plan sites in developing countries. This is mainly due to unknown boundary factors and their impact on a local production site [4]. Which leads to our research question: Which boundary factors challenge local production in least developed countries?

To answer this question, we start by explaining the advantages of a local production site and strategies of companies to enter new markets (Section 2). Subsequently, we introduce our literature research proceeding to identify production challenges in least developed countries (Section 3). The findings of this study are described as organizational, product-specific, employee-specific, technology and process-specific, purchasing and sales-specific and political challenges in section (Section 4). In order to check the topicality and practical relevance of the results found, we subsequently conduct two case studies (Section 5). We conclude with a summary of this study and future research scopes (Section 6).

## 2. Production Sites in Least Developed Countries

### 2.1. Benefits for Companies

Advantages of a local production are the overcoming of duties and the reduced product costs on site. These can be passed on to the large number of customers in these markets as these have not been developed yet. Political measures taken by governments in least developed countries to

protect their domestic markets, such as trade barriers and import taxes, are overcome by production in the target market. Other political requirements, such as local content, can also be neglected through value creation in the target market ([5] p. 193). Furthermore, companies can count on tax advantages and subsidies, which are incentives of the local government for the establishment of local corporate branches [6]. Furthermore, the product price, which is decisive in these economically strained markets, can be significantly reduced by producing locally (Figure 1). Shorter transport routes of products to the customer reduce logistics costs ([5] p. 71). Additional, low wages in developing countries have a considerable impact on production costs [6]. Components purchased locally (local sourcing) also positively influence the product price [7]. Further savings can be achieved by adapting the product to the quality demanded by the customer, which does not necessarily has to be the best possible quality, compared with conventional products [8]. In addition, lower location investments for production sites in developing countries have an indirect impact on product costs [9]. This enables an economic realization of smaller quantities at market entry and production start ([10] p. 213). With close-to-costumer production sites corporates are able to react more flexible to demand fluctuations and uncertainties [8,11,12]. This flexibility enables rapid product adjustments as they are required by the market and new products are faster on the market compared to global competitors [6,13]. Cooperation with local suppliers further increases the possibilities of flexibility, concentration on core competences and savings in purchasing [13]. Irrespective of this, the early integration of customers enables the development of cost- and function-oriented products [7,8,14]. Local production in least developed countries also has a positive influence on the manufacturer's image. This facilitates the acquisition of market shares in the respective country [11]. Figure 1 summarizes the saving options of a reference product compared to a domestic product.

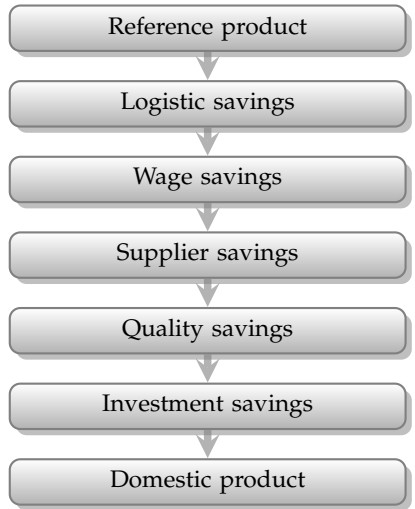

**Figure 1.** Savings through local production.

Markets of the least developed countries are characterized by a large number of new customers whose purchasing power is classified as low [15]. Nevertheless, these markets are of interest for two reasons. First, the large number of potential customers and second, the future development of these markets. Rising incomes in developing and emerging countries increase the demand of the local population for various products. According to Radjou (2012), once Maslow's basic needs are met, the focus of consumer behavior will shift [15]. This will determine the relevance of these markets for products from higher price segments in the future. Brandt and Thun (2016) currently divide demand in emerging markets into three segments [16]. The first is the premium segment, whose technically mature products are supplied by multinational companies. Second, the lower segment, whose functional and affordable products are manufactured by domestic companies. They discuss the third segment, the middle price and quality segment, which is very competitive, as domestic companies manufacture products in this category by adapting advanced technologies. Additionally,

foreign companies want to serve these large markets with adapted premium products [16]. New sales markets allow for established companies to generate increasing revenue numbers and thus company growth [17]. From a global perspective, these markets in developing and emerging countries currently drive global growth [18].

*2.2. Strategies in Least Developed Countries*

According to Leontiade (1970), to enter new markets, companies proceed according to the respective market phase of the country (Figure 2) [19]. Products are imported into the so-called *pre-markets* and less developed markets. When the markets begin to establish themselves (*take-off-markets*), the products are assembled locally and partly manufactured. In the *early-mass-markets* the local mass production is started and in the *mature-mass-markets* optimized by new technologies and local R&D. When a new market phase is reached, the price development decreases due to scaling.

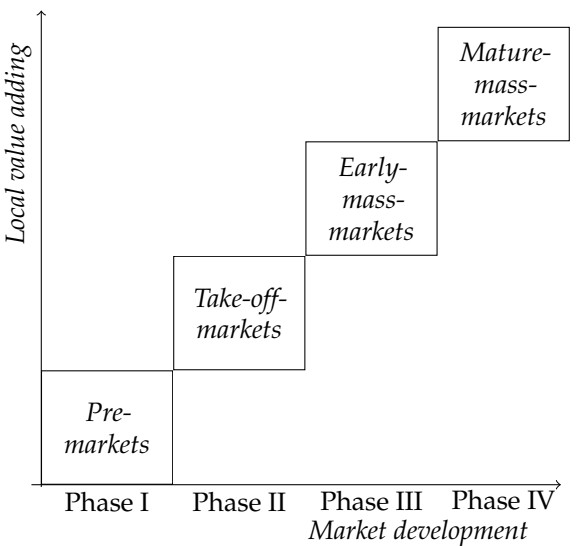

**Figure 2.** Strategies of companies in developing countries according to [19].

Putranto (2003) also describes a phase-in approach to transfer technology. Starting with (I) licensing and co-manufacturing, continued by (II) co-design and co-manufacturing and (III) joint product development least phase is a (IV) joint technology R&D [20]. Xie (2014) characterizes the different phases using the example of China. They started with the assembly line import followed by the components localization and technological adaptation and the final innovation stage [21]. A similar approach is described by Karabag (2010), who presents the case study of FIAT and TOFAS in Turkey, where the two companies aim to improve commonly quality and added value in new vehicle models [22].

Successful companies adapt their corporate and production strategies to the environmental challenges. For this reason, they analyze the past and current situation of the company to derive statements for the future ([23] p. 40). Kreikebaum (2007) defines five groups of influencing factors. First, the technological factors that determine both the product and the production processes. Second, legal conditions influencing both external and internal areas of the corporate. Economic factors are the third. They have to be considered both macro and micro economically. According to Kreikebaum (2007), the economic environment also determines the purchasing and sales market. Fourth, environmental conditions have to be considered, as well socio-cultural ones which, for example, determine the company values and their behavior ([23] pp. 41–46).

## 3. Proceeding of this paper

### 3.1. General framework

In contrast to the advantages of local production sites in the least developed countries, uncertainty about local boundary conditions pThis leads to the questions whether these boundary conditions are country-specific and whether the challenge of appropriate products for local production prevail.revent companies from creating value in developing countries ([4] p. 450). This leads to the question whether these boundary conditions are country-specific and if all corporate departments are affected when planning to produce in least developed countries. We derive the following hypotheses (H1 and H2):

**Hypothesis 1 (H1).** *The influential factors occur independently of the least developed country.*

**Hypothesis 2 (H2).** *The surrounding conditions in least developed countries impact all corporate departments.*

We want to close this research gap by a structured summary of the production challenges in the least developed countries. Therefore, we first conducted a methodological literature search. Subsequently, we present two case studies that demonstrate how these challenges affect corporates and organizations. The first case study examines a non-governmental organization (NGO) that reduces a product to its essential components and thus meets the challenges of local production. The second case study examines a small-scale car manufacturer in Tunisia, who developed his product with focus on local value adding.

### 3.2. Literature Review

According to Okoli and Schabram (2010) "a detailed methodological approach is necessary in any kind of literature research" ([24] p. 2). Therefore, we used their 8-step approach to conduct a systematic and repeatable literature review ([24] p. 7). Within this procedure the steps of data extraction and synthesis of studies are essential in order to obtain an aggregated statement about the subject area searched. Within the databases of Scopus and ScienceDirect we have searched the keywords developing countries, local production and product and process design. To expand our search, we used synonyms (1) local manufacturing and assembly, (2) Africa and India, as well as (3) product and process development. A total of 174 relevant publications were identified in the research categories of World Development, Production, Production Economics, International Business, Technology Estimation and Social Sciences. After reviewing the titles and abstracts, 73 papers were selected for final review, disregarding those not related to physical products. In addition, research, books, reviews and studies that are part of, or related to the field of process and know-how transfer research in developing countries were included in an iterative research step. The publication period was between 1970 and 2018, with the number of publications rising significantly after 2010. In the years 2017 and 2018, 11 articles were published (15% of the findings).

The results can be divided into six publication fields. In the category *technology or knowledge transfer* organizational and technological learning is investigated. *Sustainable Supply Chain* deals with the overall view of the supply chain and adjustments to customer preferences. The topics *Bottom-of-the-Pyramid* and *Frugal (Jugaad) Innovation* discuss especially customers in developing countries, in particular the development of suitable products in price, function and application for these. The research aims to offer customized products for applications and requirements in developing countries. Economic challenges to be overcome by transnational corporations and joint ventures are summarized in the category *global production*. The searched literature mentions 13 different countries within two continents as shown in Table 1. With 14 nominations (India) and 11 nominations (China), two countries are discussed significantly more often.

**Table 1.** Nominated regions.

| Region | India | China | Africa | Mexico | Brazil | Latin America | Turkey | Bangladesh |
|---|---|---|---|---|---|---|---|---|
| Findings | 14 | 11 | 8 | 4 | 4 | 4 | 3 | 3 |
| Region | South Africa | Malaysia | Kenya | South Korea | Nepal | Caribbean | Mozambique | |
| Findings | 3 | 2 | 1 | 1 | 1 | 1 | 1 | |

## 4. Production Challenges in Least Developed Countries

The challenges identified can be divided into six clusters following the systems of manufacturing companies ([25] p. 30). These are product, technology, organization, employees, sales and procurement as well as society and politics, in which a total of 97 challenges were identified (Figure 3). The results are summarized in Appendix A.

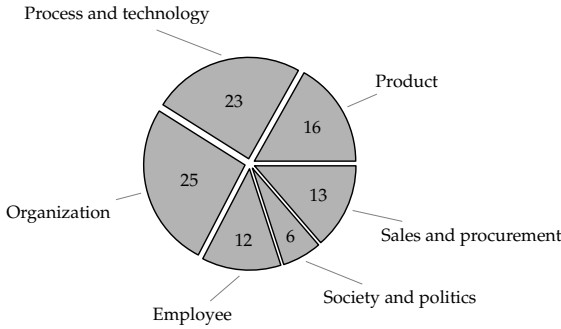

**Figure 3.** Distribution of literature findings.

### 4.1. Organizational Challenges

When setting up new production sites, various influential factors on the organization must be taken into account. E.g., the corporate culture has to enable innovation and experimentation in new markets [17] and build a corporate culture of learning and development [13]. According to Adams (2018) psychological factors such as fear of failure and fear of not being able to influence the outcome often prevent a successful start of companies in a developing country [26]. Stewart (1974) describes the difficulties in establishing a local management [27]. This is crucial as companies must build local competencies (e.g., local R&D), as they do not know the local conditions and peculiarities [28–37]. In addition, building local partnerships is seen as a success factor [38,39]. If the decision for a new location in a least developed country has been made, the duration of the production build-up is critical [40]. The distance between the parent plant and the new production site [29,31,41] and the complexity of a decentralized manufacturing network present challenges for multinational companies [42]. For this reason, the communication between the subsidies and the parent company represents a decisive barrier [43]. For production in developing countries, labor standards and conditions must be introduced and implemented according to known standards (e.g., no child labor) [44]. When the products are on the market, the infrastructure and communication channels represent a challenge for marketing, distribution and logistics [45].

The investments of multinational companies in locations in least developed countries are highly dependent on political and market stability and size [29,36,45–48]. In contrast, foreign direct investment is crucial for entrepreneurs in developing countries to achieve long-term effects [13,49]. At the same time, there are restrictions on local financing opportunities that hinder enterprises [47,50]. Due to the uncertainty in the markets and the small quantities at the beginning, companies have to wait longer until the investment is profitable. In addition, companies must expect profits that are appropriate for markets in developing countries [31,51].

When entering new markets, a company must compete with local competitors in terms of price, image and delivery time [6,11,32,44,45,52–54]. Thus the choice of the right time to enter a new market [26] and the market launch strategy is a challenge ([55] p. 162). At the same time, forecasting demand and thus production volume is difficult [31,36,47,54,56]. This is crucial because a quantities of products sold in these price-sensitive markets with low margins is essential [38,57]. While the luxury market is often occupied by foreign products, the low-cost market is addressed by local competitors, and multinational companies have to establish themselves in the middle class ahead of domestic competitors [16]. But with the low-cost variants companies run the risk that cannibalism will affect their domestic markets [53]. Local competition, know-how and protection of intellectual property pose a further challenge for companies entering a new market [5,33,54]. Also the outflow of knowledge by employees when founding spin-offs should not be underestimated [16,31,58–62]. On the organizational side, the lack of economic incentives to adapt technology and components may prevent companies from designing optimal solutions for the environment in developing countries [7,29].

### 4.2. Technological Challenges

In addition to the organizational factors, influential factors on processes and technological must be considered. According to Grant (1997), processes in developing countries should be adequate and not time-consuming [63]. Furthermore, they should be easy to build, technologically appropriate and support workers [13,29,54,64–66]. Because of the high market uncertainty, the processes should be flexible in production speed, product mix and quantities [6,52,67], have low complexity [5,38] and avoid effort [6]. Also they should be robust [68], avoid tests ([69] p. 7), quickly ramp up [70] and be standardized in processes and measurements [8,71,72]. Since the output strongly depends on the qualification level of the employees [46,54] and the technological abilities [46,65,67,73], processes should be adapted to the site-specific requirements and use labor-intensive assembly methods [64,65]. The local information system should be used [30,36,38,64] and energy saving implemented [74]. In order to achieve the required and necessary quality, corporates must stick to local production standards and practices as well as prevent waste [6,68,70]. Corporates need to establish safety standards, even if no local norms are established yet [64]. Productivity and performance is described as one of the main challenges for production sites in developing countries [75,76]. Alcorta (1999) points out high scrap rates as well as high production times and Burange (2008) mentions the lack of capacity utilization [67,77]. Referring to Terwiesch and Xu (2004) as well as Ivarsson and Alvstam (2005), a constant development and improvement of companies and technologies poses an issue for producers in developing countries [13,70]. Previously mentioned influences on the investment volume sets the challenge of low initial investment costs for technology [8,9,78]. Additionally, technology must be economic efficient for small quantities [8,54,71]. The production technology has to face demanding climate conditions in a large number of developing countries and therefore needs to be robust against environmental influences [27,29,63,79]. Oguntoye and Evans (2017) point out the importance of sustainability in technology selection [74].

### 4.3. Product Challenges

For a successful set up of new production locations in developing countries, it is crucial to offer appropriate products regarding costs, function and quality [80–85] Hereby, a reduction of complexity should be attained e.g., through reducing the number of components [86], standardization [8,16,29,64,72,87] and reduction of resources (such as financial invest, material, production time) [38,49,51]. As unappropriated products are seen as inefficient [79], local population should be involved in product development [38,79] including local product testing [79,80]. Additionally, the whole product life cycle has to be considered [79] including a plan for the product reuse [88], maintenance as well as spare parts availability [50,64,89] and low operating costs [90]. Furthermore, the product is influenced by the task of constant quality improvements [6] and assurance [5,6,50]. One of the crucial factors in developing countries are the product costs. Thomas

and Wind (2013) set the target of a competitive value, which means to meet the local populations need at the best price [91]. Consequently, low material costs are the basis [38,92]. In addition, product and processes must allow low break even points and a sustainable revenue model for small series [8,10,93].

### 4.4. Employee Challenges

The decision to produce in a developing country is accompanied by the question of the available workforce. In the literature, the lack of qualified workers is mentioned as a major challenge [5,41,64,67,94]. This lack of implicit product knowledge [7] and the different levels of education challenge production site planning [46]. The attraction of top talents and managers poses to be an immense task [30,77]. According to King (1974) as well as Bair and Gereffi (2001), the fluctuation of trained workers is challenging [95,96]. This exacerbates the lack of learning experience [43], which must be resolved by offering worker training [36,62,97]. The selection of employees must aim at building long-term relationships and trust [31,38,50,96,97]. Furthermore, the social and cultural difference, the adaptation ability of corporates to local culture and language, influences the effort involved in shifting production [41,50,64]. To support the local society corporates should strive for income equality and combat human rights violations, inequality and poverty [50,79]. Transferring the required know-how for production processes is the major task when setting up new locations [20,49,76]. Hereby, the teachability of the process, the capability of technological and market-based learning are crucial as well as a training and transfer suited for the recipient. It is therefore essential for the know-how transmitter to fully understand the process [5,40,41,71,98]. Further success factors of the knowledge transfer are the age of the technology [41,98] and the codifiability [97,98].

### 4.5. Society & Political Challenges

The product also has to be appropriate for local requirements, e.g. in security and safety, recycling and collective agreements [50,75]). Furthermore, import taxes, tariff and non-tariff trade barriers, local content requirements and financial restrictions challenge corporates [5–7,11,67].Additional politics in developing countries demand social commitment of corporates (e.g. create jobs, increase GDP and combat poverty) [6,38,42,78,79].By contrast some countries are characterized by the lack of political support [48], corruption, abuse and theft [36,47–49,54,56] as well as political instability and high inflation rates [67].

### 4.6. Sales & Procurement Challenges

With local suppliers often being required by political restrictions, they are also indispensable for achieving cost targets [13,27,28,50,64]. Obstacles regarding local suppliers occur because of quality issues, unreliable delivery times, high communication costs, the lack of transport routes and the company's size [48]. Further literature mentions the lack of flexibility and innovation culture on the supplier side [7,38,99]. Because of this, the MNCs (Multi National Cooperation) must provide supplier training or assistance (e.g., for standardization) [33,100], build a collaborative relationship [13,34,76,100] and integrate the supplier into their existing network [91]. Local suppliers influence the product, therefore it is recommended to use modules from established [7] and standardized suppliers [33,48,80,100]. The technology choice is highly dependent on the technological capabilities of the supplier. Therefore, technological development of the whole supply chain is necessary despite high costs and expenses [7,73] to secure the quality of vendor components [48]. The lack of an expanded infrastructure has a major impact on the expense of logistics resulting in high distribution costs [29,30,75]. Karamchandani et al. (2011) discuss the underdeveloped economic ecosystems [53]. Building on this, several studies recommend the development of clusters to gain synergies [33,71,96,101,102]. Further, the lack of a reliable infrastructure that would enable the aspried customer proxity posses to be difficult [5,6,50,89,92].

*4.7. Conclusion of the Literature Research*

A number of 97 challenges occurring in developing country while setting up new production locations or maintaining production sites are described in the current literature. The division into six clusters enables an overview. However, these categories are by no means to be regarded as isolated but are much more in context and interrelate with each other. For example, the infrastructure is directly related to the supplier structures [103]. The number of nominations depends on the concentration of authors within their publication on a region and not on their attempt to present the challenges holistically. This confirms H1, that the influential factors are not country-specific, but rather depend on the level of development of the countries. The categories indicate that from the organization and management structure, integrated product and production development to marketing, sales and legal, all business units must be adapted for local production. This confirms our second hypothesis (H2) and highlights the need for integrated planning of local production sites for long-term success.

## 5. Case Studies

In order to prove the actuality of the challenges we conducted two case studies, which also demonstrate how to handle these challenges successfully with low monetary expense. These case studies are conducted using the procedure of Yin (2018), and follow the steps of theory development, case selection, reporting and cross-case conclusion ([104] p. 49). The studies are intended to demonstrate the relevance of the challenges identified previously. In addition, the cases should also demonstrate the importance of global supply chains through their supply structure. The selection of the case studies was based on the products developed specifically for the environmental conditions and their successful establishment in the local markets. Furthermore, theses cases fulfill the requirement of documenting extreme cases [105], due to their financial constraints. Extreme cases are used because of their special adaptation to the respective situation [105]. The data was gathered using semi-structured interviews and secondary data. The *OneDollarGlasses* project was founded in 2012, started training in Africa in 2013 and obtains its raw materials from Germany and China. *Wallys* was founded in 2006 and has since produced more than 2200 vehicles with a PSA engine. The case study data was aggregated through interviews and secondary data.

*5.1. "OneDollarGlasses"*

The aim of the project *OneDollarGlasses* is to provide people worldwide durable and locally produced eye-wear. The project is being carried out by the association EinDollarBrille e.V. with its headquarter in Erlangen (Germany). Approximately 400 people work for the organization. Half of these work in Germany and Switzerland on a voluntary basis, the other half are permanently employed in eight African countries, India and Latin America (https://www.onedollarglasses.org/).

The *OneDollarGlasses* consist of three components: a lightweight, flexible and sturdy spring steel frame, poly-carbonate lenses and heat shrink tubing for the eyeglasses (Figure 4). Due to this structure, the material costs are about 1 USD. The selling price is region-specific at approximately two to three times the normal daily wages. The frame, which is available in three different sizes, is manufactured on a bending machine which fits into a wooden box of $30 \times 30 \times 30$ cm. With one machine and six producers, an annual volume of 50,000 glasses can be produced. The diopter spectrum of *OneDollarGlasses* ranges from $-10.0$ to $+8.0$ diopters, which are available in 0.25 diopter steps. With three frame sizes and 38 variants of lenses per side, 4332 different glasses can be customized. The lenses are mounted by simply clicking them into the spring steel frame. To customize the glasses, opticians visit the villages and assemble the glasses on site. The aim of the project is sustainability through profitability, which means that people can live from the production and sale of One Dollar Glasses. Thus, the *Social Business concept* is located in the fields of health, education and social/economic development. The quality is ensured by a multi-level system, as samples of the local produced glasses and the materials are additionally tested by employees from Germany.

The selling price is secured by documentation in the patient's books. Furthermore, the supply of materials is discontinued when employees leave the organization, ensuring that no glasses are sold by former employees.

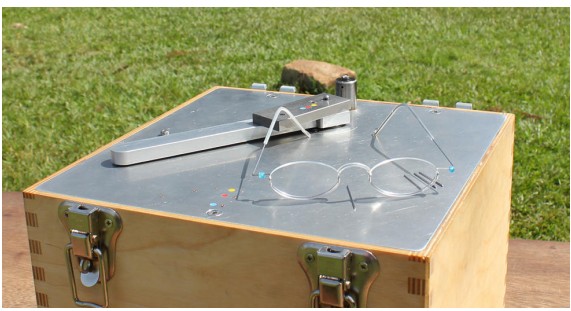

**Figure 4.** *OneDollarGlasses* and bending machine (2012).

*5.2. Overcoming the "OneDollarGlasses"-Challenges*

Of the aforementioned production challenges in least developed countries, we select 12 challenges to demonstrate how they were solved in the *OneDollarGlasses* case study (Table 2).

**Table 2.** Challenges and solutions: *OneDollarGlasses*.

| Challenges | Solution |
|---|---|
| *Product challenges* | |
| Reduction of complexity | Standardized frame and lenses |
| Reduction of components and resources | Only essential components |
| Standardization | Low internal product variety |
| *Employee challenges* | |
| Lack of qualified workers | Customized training concept, open source manual |
| Combat poverty | |
| Know-how protection | Controlled material supply |
| *Technological challenges* | |
| Low process complexity | Customization during assembly, portable equipment |
| Low investments | No automation |
| Quality standards | Templates for quality assurance |
| *Sales and procurement challenges* | |
| Standardized suppliers | Central purchasing to enable economies of scale |
| Quality assurance | Central quality assurance |
| Lack of Infrastructure | On-site assembly and direct sales |

By reducing the glasses to essential components, it is possible to manufacture the glasses locally and at the stated price (Figure 4). The system of easy-to-install lenses allows customization. Through these steps, the glasses overcome the challenge of the adapted product in terms of function and price, which is feasible for the customer. With three frame sizes and 25 lens types, glasses are strongly standardized, and concurrently customer-specific. Customer proximity is achieved by opticians who assemble and sell the glasses on-site. The challenge of quality assurance is solved by inspecting the material in Germany before shipping it to Africa. The controlled material flow also prevents the product copying and thus secures the know-how in the long term.

The training of the local employees is carried out by specialists with a customized training concept. Local employees generate jobs and combat prevailing unemployment and poverty. By reducing the manufacturing process to the essential, investment costs are low. Using an aggregate procurement managed by the association, economies of scale are enabled. Furthermore, the controlled incoming goods inspection within the procurement process enables high quality product standards. Challenging infrastructure is tackled in two ways. First, using a local production process which is enabled by the

portable production equipment. Second, the direct sales concept allows customer contact even in inoperable terrain and widely dispersed customers.

### 5.3. Wallys

Founded in 2006, *Wallys* is a family-owned Tunisian automotive manufacturer with approximately 120 employees. Of these, 40 work directly in administration, development and production at the main plant in Tunis. In 2008, *Wallys* presented their first prototype at the Paris Motor Show, started production in 2009 and released the second model in 2013. With 350 vehicles per year and more than 2200 vehicles in total, *Wallys* is one of the largest domestic manufacturer of vehicles in Tunisia. *Wallys* aims to build reliable and safe vehicles (https://www.wallyscar.org/).

The *IRIS* (Figure 5) is a convertible sport utility vehicle and built for daily use and leisure. The vehicle is maneuverable and easy to park. With a top speed of 139 km/h and an acceleration (0–100 km/h) of 11.2 s, it satisfies the customer needs. Starting at 12.500 Euro, an extensive option list for additional features, like ABS, is available. The vehicle itself consists of a bent and welded main frame made from galvanized steel, making the vehicle suitable for challenging conditions. In this main frame the drive and chassis components are mounted, whereby the engine is supplied by the PSA group. Next, the bodywork, manufactured by a supplier and made of fiberglass, is attached. The production steps of IRIS are designed for manual work, which enables the customization of almost every component.

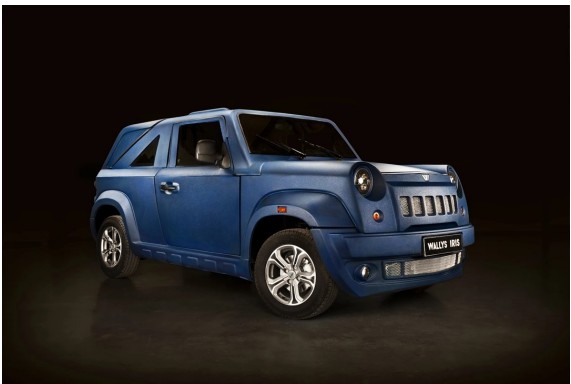

**Figure 5.** The Iris, presented 2013.

### 5.4. Overcoming the "Wallys"-Challenges

In this case study we describe how the selected challenges were handled by a small series vehicle manufacturer (Table 3). *Wallys* decided to use fiberglass in the bodywork which allows for low investment costs and hence the vehicle price. Additional, this technology allows for outsourcing, since the processing complexity is not high compared to deep-drawn components. The decision not to use curved windows is also based on the available technology. According to the founders of *Wallys*, curved windows cannot be supplied by any Tunisian company. With the focus on manual work in manufacturing as well as final assembly, no major investments in automation are necessary. To design the ideal vehicle for the customer, *Wallys* relies on local R&D. During the vehicle development, the focus was vehicle robustness and the possibility of individualization by the customer. To control production complexity the chassis, bodywork and wheels are standardized.

Particularly challenging for *Wallys* in the category of employees is the lack of product knowledge among new employees and the high staff turnover, which is approximately 30% per year. Because of this, the founders rely on intensive in-house training of new employees to qualify them for their work tasks and bring them closer to the company culture, including necessary time for training and learning. The company culture also includes continuous improvement, which was exemplary explained in the interview by the example of the vehicle's doors. With these, the gap dimension was successively reduced in many iterations and the processes were adjusted to the smaller tolerances. A major challenge

**Table 3.** Challenges and solutions: *Wallys*.

| Challenges | Solution |
|---|---|
| *Product challenges* | |
| Reduction of complexity | Straight windshields, fiberglass bodywork |
| Reduction of components and resources | Continuous improvement |
| Lack of local requirements | Homologation in Europe |
| Standardization | Standard components like the drivetrain |
| *Employee challenges* | |
| Lack of qualified workers | Intensive inhouse training, manual work tasks |
| High fluctuation of qualified employees | Company culture, intensive inhouse training |
| *Technological challenges* | |
| Low process complexity | Concentration on assembly |
| Low investments | No automation |
| Manual work tasks | Components especially designed for manual assembly |
| *Sales and procurement challenges* | |
| Local suppliers | Building and maintaining a local supplier base |
| Collaborative relationship | Early integration of suppliers |
| Unreliability of suppliers | Supplier quality management |

during vehicle development was to achieve local homologation. The local requirements were not specified and communicated, which is why *Wallys* concentrated on a homologation in Europe.

In procurement, local suppliers are the main issue. The supplier quality, reliability and delivery time are unpredictable. This is why *Wallys* relies on a well-founded selection of longterm suppliers. In addition, *Wallys* aims to integrate suppliers at an early stage. If this is not possible, a quality management employee for suppliers ensures reliability. Several sources are also used for core components in order to avoid supply bottlenecks. In the sales sector, after-sales management is difficult due to the dispersed customer base. Therefore, *Wallys* designed his vehicle so that repairs could be carried out by the customer himself or by suppliers like the PSA Group.

## 6. Conclusions

In summary, by conducting a literature research, we identified in total 97 production challenges in developing countries. These are aggregated from the literature sources and are partly found in related studies [5,16,106]. Case studies of independent products and countries show the actuality of production-related influences in least developed countries. They also demonstrate ability to overcome these challenges by product development (e.g. reduction of production complexity of the glasses), technology (e.g. using simple process-able fiberglass for the bodywork) or organizational measures (e.g. central quality assurance of procured material for the glasses frames). Our study is still subject to limitations: The literature study is limited by the choice of search terms and the databases used. In addition, only English literature was searched. The case studies show that it is not the monetary expenditure that is decisive in overcoming the challenges. Nevertheless, especially small companies may face specific problems that are not relevant for MNC in practice and therefore are not revealed by our study. Future research should examine the requirements arising from these boundary conditions (in line with [107]) and changes in the challenges related to the development of these countries. A comparison between the challenges faced by the least developed countries and those faced by the developed countries would further detail the least developed country-specific factors. The definition of least developed country as well as examples to overcome these should encourage companies to plan and establish sustainable and local production sites in least developed countries.

**Author Contributions:** Conceptualization, M.B.; methodology, M.B.; investigation, S.S. and M.B.; writing–original draft preparation, M.B.; writing–review and editing, M.B. and M.L.; visualization, S.S. and M.B.; supervision, M.L. All authors have read and agreed to the published version of the manuscript.

**Funding:** This research received no external funding.

**Acknowledgments:** Lienkamp gave final approval of the version to be published and agrees to all aspects of the work. As a guarantor, he accepts responsibility for the overall integrity of the paper.

**Conflicts of Interest:** The authors declare no conflict of interest.

## Abbreviations

The following abbreviations are used in this manuscript:

GDP     Gross Domestic Product
MNC     Multi National Corporation
R&D     Research and Development

## Appendix A

**Table A1.** Literature research.

| Cluster | Category | Challenge | ∑ | Source |
|---|---|---|---|---|
| Organization | Corporate | Corporate culture | 2 | [13,17] |
| | | Psychological fear | 1 | [26] |
| | | Local corporate management | 1 | [27] |
| | | Local competency | 11 | [28–37,77] |
| | | Local partnerships | 2 | [38,39] |
| | | Set up duration | 1 | [40] |
| | | Distance to main plant | 3 | [29,31,41] |
| | | Complexity of decentralization | 1 | [42] |
| | | Communication between subsidies and parents | 1 | [43] |
| | | Local labor standards | 1 | [44] |
| | | Aftersales processes | 1 | [45] |
| | Investment | Investment climate | 8 | [29,36,42,45–48,66] |
| | | FDI and longterm effects | 2 | [13,49] |
| | | Local financing | 3 | [47,50,108] |
| | | Appropriate profits | 3 | [31,51,77] |
| | Strategy | Local competitors | 8 | [6,11,32,44,45,52–54] |
| | | Time of market entry | 2 | [5,26] |
| | | Time-to-Market | 2 | [6,55] |
| | | Uncertainty of demand | 5 | [31,36,47,54,56] |
| | | Small margins | 2 | [38,57] |
| | | Market segment | 1 | [16] |
| | | Cannibalism effect in the home market | 1 | [53] |
| | | Intellectual property rights | 3 | [5,33,54] |
| | | Spin-offs | 8 | [16,31,58–62,109] |
| | | Economic incentives to adapt products and technology | 2 | [7,29] |

**Table A1.** *Cont.*

| Cluster | Category | Challenge | Σ | Source |
|---|---|---|---|---|
| Technology | Process | Adequate and low in expedience | 8 | [13,29,54,63–66,83] |
| | | Flexible in speed, product mix and quantities | 5 | [6,52,55,67,110] |
| | | Low in complexity | 2 | [5,38] |
| | | Low in effort | 1 | [6] |
| | | Robust | 1 | [68] |
| | | Avoid tests | 1 | [69] |
| | | Fast to ramp up | 1 | [70] |
| | | Standardized in process and measurement | 8 | [8,30,34,41,60,64,71,72] |
| | | Output depends on the worker's skill level | 3 | [46,54,111] |
| | | Output depends on technological capabilities | 4 | [46,65,67,73] |
| | | Appropriate to local requirements and assembly intensive | 9 | [5,27,29,47,54,57,64,65,90] |
| | | Local information system | 4 | [30,36,38,64] |
| | | Energy saving | 1 | [74] |
| | | Local manufacturing norms and waste reduction | 12 | [6,32,47,49,64,66,68,70,82,89, 110,112] |
| | | Work safety | 1 | [64] |
| | Productivity | Productivity and performance | 11 | [5,27,35,47,66,71,75–77,82, 113] |
| | | High scrap rates | 1 | [67] |
| | | Lack of capacity utilization | 1 | [77] |
| | | Constant development and improvement | 2 | [13,70] |
| | Investment | Low initial investment | 8 | [8,9,29,39,75,78,90,114] |
| | | Economic efficiency for small-scales | 3 | [8,54,71] |
| | Environment | Robust against environmental influences | 4 | [27,29,63,79] |
| | | Sustainability in technology selection | 1 | [74] |
| Product | Components | Appropriate in costs, function and quality | 27 | [7,16,27,29–32,36,39,42,45,47, 50,52–54,57,61,78,80–85,91, 115] |
| | | Reduction of components | 1 | [86] |
| | | Standardization of components | 6 | [8,16,29,64,72,87] |
| | | Reduction of resources (financial, material, time) | 9 | [26,27,29,30,38,47,49,51,114] |
| | | Involvement of local population | 2 | [38,79] |
| | | Local testing | 2 | [79,80] |
| | | Consideration of life cycle | 1 | [79] |
| | | Plan for reuse | 1 | [88] |

**Table A1.** *Cont.*

| Cluster | Category | Challenge | Σ | Source |
|---|---|---|---|---|
| | | Maintenance and spare parts availability | 9 | [27,29,50,58,64,73,89,90,114] |
| | | Constant quality improvement | 1 | [6] |
| | | Quality assurance | 8 | [3,5,6,29,50,58,67,71] |
| | Costs | Competitive value | 1 | [91] |
| | | Low material costs | 2 | [38,92] |
| | | Low break-even points | 7 | [8,10,27,29,38,57,93] |
| | | Economics of scale and scope | 5 | [7,27,32,38,54] |
| Employee | Qualification | Lack of skilled employees | 27 | [5,26,27,30,31,33–35,37,39,41, 47,50,54,56,58,61,62,64,65,67, 73,94,97,114,116] |
| | | Lack of implicit product knowledge | 1 | [7] |
| | | Different education level in different DCs | 1 | [46] |
| | | Attraction of top talents and managers | 2 | [30,77] |
| | | Fluctuation of trained employees | 2 | [95,96] |
| | | Lack of learning experience | 1 | [43] |
| | | Offering worker trainings | 4 | [36,62,83,97] |
| | | Focus on long term relationship with employees | 5 | [31,38,50,96,97] |
| | | Integration of local culture and language | 11 | [26,29,34,36,39,41,50,56,64, 76,109] |
| | | Income equality, combat human rights violations, inequality and poverty | 8 | [26,27,38,50,78,79,96,109] |
| | Know-how | Know-how-transfer | 18 | [20,32–35,37,43,49,54,58,61, 62,73,76,80,83,87,117] |
| | | Know-how transmitter | 7 | [5,40,41,41,58,71,98] |
| | | Age of technology | 2 | [41,98] |
| | | Codifiability | 2 | [97,98] |
| Society & Politics | Politics | Local requirements (e.g., safety, security, recycling) | 19 | [3,6,16,28,32,36,38,39,42,45, 50,54,56,58,62,75,80,82,108] |
| | | Import taxes, trade barriers, local content, financial restrictions | 18 | [3,5–8,11,13,14,27,29,58,66, 67,71,73,78,82,90] |
| | | Social commitment | 8 | [6,31,38,42,66,78,79,89] |
| | | Lack of political support | 1 | [48] |
| | | Corruption, abuse, theft | 7 | [36,47–49,54,56,109] |
| | | Instability and high inflation rates | 1 | [67] |
| Sales & Procurements | Supplier | Requirement of local suppliers | 5 | [13,27,28,50,64] |
| | | Quality issues | 1 | [48] |
| | | Unreliable delivery times, high communication costs, inflexibility and lack of innovation culture | 21 | [7,29,32–34,38,39,47,52,55,57, 58,62,71,73,80,84,90,99,110, 113] |
| | | Supplier training | 2 | [33,100] |
| | | Collaborative partnership | 4 | [13,34,76,100] |

**Table A1.** *Cont.*

| Cluster | Category | Challenge | Σ | Source |
|---|---|---|---|---|
| | | Integration of suppliers into network | 1 | [91] |
| | Product | Usage of modules | 1 | [7] |
| | | Standardized suppliers | 4 | [33,48,80,100] |
| | Technology | Technological capabilities of supplier | 2 | [7,73] |
| | Infrastructure | Logistic/Distribution costs | 10 | [29–32,39,48,52,53,57,75] |
| | | Underdeveloped business ecosystem | 1 | [53] |
| | | Building clusters | 5 | [33,71,96,101,102] |
| | | Lack of reliable infrastructure | 5 | [5,6,50,89,92] |

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
