# Peer review of "Production Challenges in Least Developed Countries"

_challenges, doi:10.3390/challe11010001_

Round 1

Reviewer 1 Report

This study addresses challenges that local production by multinational corporations in less developed countries faces. The paper suggests the factors found through literature review and the results of the two case studies. Despite its extensive approach, the connection between literature review and case studies looks weak. Do the case studies examine the influence of organizational, technological, product-related, employee-related, socio-political factors on local production? How does creating the list of 97 challenges gain legitimacy? The result of literature research needs the rationale behind the categorization. How can the authors say simply that the solutions soothe the specified challenges?  The term "influences" is misleading. Do the authors mean challenges or influential factors? In the very beginning, I suggest the paper should start with relevant research questions. 

Reviewer 2 Report

The study contributes to organizational, product-specific, and technological solutions to meet the challenges in the case of developing countries. The Authors also characterize two examples of their approach to challenges assessment in the case of the production area.

I recommend to Authors to add the directly defined aim of the paper in the abstract part.

It will help the reader to settle the area of research from the beginning.

The aim also does not appear in the case of the introduction.

It will be also beneficial for the further reader to set the plan of the article at the end of the introduction.

The Authors did an extensive and broad literature review with is, for sure, the strong part of the study – especially the table part of it.

Based on study literature, the Authors could state some hypothesis which would clear the scientific aim of this literature background. I have no objections to a literature review.

I am happy that the Authors underlined the scope of further research. However, they should also add the limitation of the study.

Authors recognize the 97 challenges occurring in developing countries in case of planning new production locations. However, if there is similar research that would include developed countries and the number of challenges?

In part 5

I suggest adding 2-3 justification sentences to choose precisely such cases.

Line 328

The case studies cover companies that are not financially comparable to MNCs. Nevertheless, they show that it is not the financial means that are decisive but the idea to overcome.

Do other studies also confirm this conclusion?

Reviewer 3 Report

The novelty of the article is based on interesting topic and actual theme. The paper aims to describe and analyse the production challenges in developed countries. The literature analysis is sufficient and provides grounded results. Still it remains not very clear why are selected very different examples of good practices except they quite well represent distinguished characteristics important entering the market at developing countries.   

Round 2

Reviewer 1 Report

I still don't understand theoretical implications of this study, and the study is methodologically weak. 97 challenges are not derived from a scientific method, and the case studies did not address why the cases are important and deserve attention. 

Author Response

Dear reviewer,

as mentioned, we base our research approach on the approach of Okoli and Shabram. According to GoogleScholar this paragraph was used by 879 other studies. This method is used to identify the 96 challenges. The procedure to support literature reviews with actual case studies is also a well-known procedure.

According to Yin (1994) case studies are selected to describe extreme scenarios: In our study: the case studies represent use cases that show that not the monetary expenditure but the company and situation-specific adjustments are decisive for success. They are extreme in the kind of optimizing the product and manufacturing process. The glasses are simplified and optimized to their essential components. The car as a complex product with many components demonstrates that also supplier integration is crucial.

What do you suggest in order to optimize the weaknesses that exist from your point of view?

Kind regards

Matthias Brönner